# Gulliver and the Rabbis: Counterfactual Truth in Science and the Talmud

**Menachem Fisch**

The Cohn Institute for History and Philosophy of Science and Ideas, Tel Aviv University, Tel Aviv 6997801, Israel; fisch@tauex.tau.ac.il

**Abstract:** The paper presents Jonathan Swift's *Gulliver's Travels* as the first systematic attempt to claim that the normal methods of testing belief and opinion for clarity, consistence, coherence, and how they stand to the facts are powerless when applied to deep-seated normative commitments, or what Wittgenstein dubbed "framework truths." To subject our norms to normative critique requires a measure of self-alienation that cannot be achieved merely by looking hard at or thinking hard about our world and ourselves. However, by closely examining the contrived counterfactual scenarios (or, as I have shown in former work, by exposure to the normative critique of significant others), that Swift is shown to claim, such normative framework assumptions can be challenged to great effect! The standard epistemologies of his day—Baconian empiricism and Cartesian rationalism—fiercely ridiculed in the course of Gulliver's third voyage are cruelly dismissed as powerless to change the course of science and keep it in normative check. The transformative effect of the clever thought experiments presented in the three other voyages (of imagining London shrunk to a twelfth of its size and enlarged to giant proportions, and a more responsible and intelligent race of beings inserted above (normally sized) humans) enable Swift to obtain critical normative distance from several major assumptions about politics, religion, aesthetics, ethics, and much more, including the limits of the thought experiment itself. The paper then goes to show how the same kind of counterfactual scenarios are put to impressive use in the Talmudic literature, with special reference to foundational questions of ethics and law.

**Keywords:** thought experiments; framework assumptions; Jonathan Swift; Newtonian Physics; Tamudic law of lost property; Talmudic laws of Damages; profanation of God's Name

## 1. Introduction

The idea that science owes much of its epistemic authority to the empirical testability of its theories runs deep. Insofar as science aims at adequately describing and explaining the world we experience, it is to the tribunal of the facts of our sense experience (to borrow Quine's term[1]) that its efforts should be held accountable. Whether or not there are grounds for claiming that the facts we experience represent, correspond, or in some sense resemble the world without, is a different and disputed question that has no bearing on the empirical nature of scientific authority. (I, for one, side with those who consider it a question that cannot even be posed coherently, but that is beside the point.) It is against this backdrop that scientific thought experiments emerge as so puzzling.[2] Regardless of the many questions concerning the structure and inner logic of thought experiments, it is the idea

---

[1]  W. V. Quine (1951, p. 41).
[2]  For a recent and comprehensive discussion of thought experiments and the problems they raise, see: Stuart et al. (2018), especially Parts Three and Four.

that studying imaginatively contrived and especially counterfactual scenarios can somehow teach us important lessons about the world we live in that renders them so baffling. How can what is patently *not* the case teach us anything about what it is?

When thinking of thought experiments in science, one is naturally drawn to the well-known, highly imaginative transformative scenarios of Galileo's frictionless lake and planes, Maxwell's demon, or Einstein's rideable light ray. Needless to say that the employment of thought experiments is by no means confined to science. However, the first person in the modern era who seems to have believed that the close study of counterfactual scenarios is the *only* reliable way to challenge well-entrenched normative commitments and develop fundamentally new insights in and outside science, was, I believe, Jonathan Swift, of all people. A word of introduction is in order.

## 2. The Philosophical Conundrum of Newtonian Physics

Swift's *Gulliver's Travels* was published in 1726, less than a year before the death of Isaac Newton. It was the heyday of Newtonianism, and his 1687 *Philosophia Naturalis Principia Mathematica* ruled supreme as an unprecedented scientific achievement.[3] Yet without in any way detracting from its scientific merit, and in a sense because of it, Newton's *Principia* and subsequent *Optics* proved to be philosophical nightmares whose impact on philosophy of science, as I have remarked elsewhere, was to transform its very nature.[4]

Prior to Newton, Western philosophy was primarily concerned with the constitutive project of properly *establishing* science anew. Francis Bacon's *Novum Organum* (1620) and René Descartes' *Discourse on Method* (1637) and *Meditations* (1641) stand out in this respect. Each presented a radical appeal to begin the study of nature from scratch, and to establish it de novo upon sound (yet profoundly different) epistemological and methodological grounds. Bacon's Protestant Reformation philosophy insisted on grounding scientific truth in the immediacy of unquestionable factual experience from which one was then to reason *up* inductively to theory—a position adopted by the newly-founded Royal Society as its 'official' philosophy[5]. Descartes's Catholic Counter-Reformation philosophy insisted, by contrast, that it be grounded in the a priori certitude of mathematical truth given by pure reason, from which one then reasons *down* deductively to the facts.[6] Both philosophies promised lofty, ambitious, and religiously meaningful visions of grand scientific accomplishment, backed by foundational and systematic philosophical reasoning. They also contradicted each other completely!

The problem with the *Principia* was that it exhibited a perplexing mixture of mutually irreducible empiricist and rationalist elements, a combination that neither philosophy was capable of accommodating from its own purist perspective. Normally, when confident subscribers to a well-established philosophy of science find a work of science to rudely defy their prescriptions, they will tend to declare it confused or incoherent.[7] But in the case of Newton's great work, such a response was out of the question. Its unanimous reception as an unprecedented scientific accomplishment rendered it virtually immune to philosophical criticism and ruled out any possibility of such philosophic

---

[3]   Alexander Pope's famous epitaph for Newton says it all: "Nature and Nature's laws lay hid in night: God said, Let Newton be! and all was light". However, unlike Kant, who viewed Newtonian physics as a solid and irreversible if intermediary step in a convergent sequence directed by the "regulative principles" of reason, Swift (putting the words in the mouth of Aristotle's ghost during Gulliver's visit to Glubbdubdrib in the course of his third voyage) wisely declares that even the sciences of "Attraction" (gravitation), "whereof the present Learned are such zealous Asserters" will eventually too "be equally exploded", deeming "new Systems of Nature" to be "but new Fashions, which would vary in every Age; and even those who pretend to demonstrate them from Mathematical Principles, would flourish but a short Period of Time, and be out of Vogue when that was determined" (Bk.III, chp. 8, §2).

[4]   See Fisch (2008). The following paragraph is based on pages 526–27 of that paper.

[5]   The front piece of Thomas Sprat's 1667, *The History of the Royal Society* provides one of many avid contemporary expressions of the Society's commitment to Bacon.

[6]   On the religious origins and orientations of the two philosophies see the paper mentioned in the previous note. On the Protestant literalist roots of Baconian empiricism see also MacKean (1987, p. 75), and on the Catholic roots of Descartes's rationalism, Toulmin (1991) especially chp. 2–3, and Menn (1998, p. 53ff.).

[7]   And explained away as the sorry, irrational or a-rational result of 'external' circumstance. See Lakatos's (1978) infamous.

reproach. Both philosophies of science were henceforth forced to rethink their positions in the light of the new physics[8], and the philosophical project of *establishing* science was rendered obsolete. From that point on philosophy of science was rendered an essentially *interpretative* rather than *constitutive* undertaking. If for Bacon and Descartes the task of philosophy was to legislate epistemologically and methodologically for a reliable interpretation of *nature*, after Newton (and famously for Kant), the question for European philosophy of science became that of making philosophical sense of the particular interpretation of nature that Newton had fashioned.[9]

At the time Swift was writing *Gulliver*, the problem of accounting philosophically for the new science's unprecedented achievement was already acutely felt, but there was no viable solution yet in sight. Bacon's insistence that scientific inquiry be consciously shielded from what he insisted was the inevitably distortive contribution of the ever-prejudiced mind and that it should proceed by "inducing" nature to freely speak for herself, as it were, deemed any talk of imaginatively conjectured—let alone counterfactual scenarios—as severely detrimental to scientific research. Bacon's experimental methodology required ingenuity and cunning in establishing experimental and protocols but never in mindful attempts to "anticipate" nature, as he put it. Cartesian rationalism was not better off in this regard. Rationalists indeed maintained that mind alone, never the senses, was capable of vouching for the kind of mathematical certainty deserving of science, which, for them too, deemed the very notion of scientifically meaningful *hypothetical* reasoning a contradiction in terms. The former envisaged science-at-its-best as yielded by an elaborate, 'literalist' process of reading the truth *out*, or *off* the empirical data; the latter, envisaged it as yielded by an equally elaborate process of reading the truths of mathematics *in* to what meets the eye. By the time Swift was writing *Gulliver*, as noted at the outset, science-at-its-best was no longer an abstract philosophical promise but an astounding reality that the two grand and theologically inspired philosophies that had promised it were utterly powerless to explain!

## 3. Gulliver's Travels Philosophical Project

Swift's cruel, Aristophanian ridicule of both positions in the course of Gulliver's third voyage, first to the flying island of Laputa, then to Balnibarbri, and then, to a lesser extent, to the ghosts of Glubbdubdrib, is noted in the literature, but nowhere is it properly contextualized. Philosophers of science have failed to pay due attention to the book. The Royal Society, rather than Bacon's philosophy proper, is widely taken to be the satirized target of the pointless and unguided experimental projects of Balnibarbi's "Academy of Projectors". But preciously few have interpreted the floating Island of Laputa, with its wholly impractical other-worldly obsession with mathematics, as Balnibarbi's knowingly Cartesian doppelgänger. Anne Mulhall comes close in characterizing Laputa as a ridicule of the rationalist's autistic, dysfunctional preoccupation with the "life of the mind"[10]. But by focusing on the "language debates" of the time rather than on epistemology, the rationalism her otherwise excellent paper takes to be Swift's Laputian target is the linguistic opposite of nominalism, rather the epistemological opposite of Baconian inductivism.

---

[8] Apart from its incapacity to accommodate the more inductivist and empirical portions of Newton's physics (in particular Book III of the *Principia*, and the "proofs by experiment" systematically employed throughout the *Optiks*, let alone the "Rules of Reasoning in Philosophy [i.e., Science]" situated between the *Principia*'s Books II and III), Descartes rationalism faced insurmountable difficulties accounting for the differential and integral calculus fashioned by Newton and Leibniz that mathematically grounded Newtonian physics. As Berkeley's *The Analyst* of 1734 amply proved, the calculus, despite the indispensable role it played in Newton's physics, fell considerably short of the standards of mathematical self-evident certitude (clarity and distinctness) required by Descartes' philosophy.

[9] The interpretative turn in philosophy of science reached completion in William Whewell's monumental *Philosophy of the Inductive Sciences* of 1840 significantly subtitled '*Founded on their History*', referring to his three-tome *History of the Inductive Sciences from the Earliest to the Present Time* published three years earlier.

[10] Mulhall (2012, pp. 35–36).

However, the vast majority of readers fall even short of that. For some reason, many take Laputa to be a spoof of Bacon's New Atlantis[11], which can only apply if it is divorced from the structure and methodology of Bacon's original experimentalist vision of Salomon's House as well as made to stand in for a spoof of the science of Swift's day[12] or, as one recent critic proposes, the developing scien*tism* of his day.[13] It is no wonder, therefore, that several readers of Gulliver's third voyage view it as "tangential" to, and its literary and satirical merits to fall short of, those of the other three.[14] The fact that Book Three was written last militates against such a position. Swift seems to have considered the third voyage, if not the work's crowning achievement, then at least the element that lent it coherence and rendered it whole.

If the target of Gulliver's visits to Laputa and Balnibarbi was science or scientism—i.e., science's remoteness or irrelevance to the important questions of politics, human welfare, and religion—it is unclear why the mathematical and empirical sciences were relegated to two so very different locales and why the science practiced in them failed not resemble Newtonian theory or practice in the least. Moreover, the time bound ever-changing and open-ended view of science conveyed to Gulliver by the ghosts of Aristotle, Gassendi, and Descartes in Glubbdubdrib can on no account be seen as expressing dissatisfaction with science, only of the conceit of viewing its current achievements as final. Add to that the scientific (or science-fiction) marvel of the man-made floating Island of Laputa (a technological fantasy woven of Gassendian and Newtonian themes), which, despite the ugly and violent use to which it is put, is not condemned for being scientific, but for the immoral abuse of its power.

All this points to one conclusion: That what Swift so bitterly satirizes in Laputa and Balnibarbi is not the science of his day, but the two foundational philosophies that lamely boasted to ground, explained it, and provided the foolproof methodology for its future development. It is not the *body* of contemporary scientific knowledge that is under attack in the third voyage but the two contrasting *images* of knowledge, to use Elkana's well-known terms, that each alleged to constitute science's normative framework.[15]

As noted above, the slapstick ridicule of the two philosophies in Book Three resembles Aristophanes's lampooning of intellectual fashions in "Clouds". For all its sarcastic mockery, "Clouds" bears no real philosophical significance. Nor would have Swift's foray into epistemology, had it remained confined to the negative derision of Laputian rationalism and Balnibarbian empiricism. However, and this is my main claim with respect to the book, quite unlike Aristophanes, Swift's ridicule of the two philosophies' does not stop at that but comes accompanied by an intriguing epistemic and methodological alternative vividly exemplified in Gulliver's three other voyages.

## 4. Counterfactual Scenarios

New fundamental truths about the world we live in and new challenges to what we thought were fundamental truths, argues Swift by extended example, are achieved not by amassing data and aimlessly casting around for new phenomena or by devising and applying new mathematical formulae. New knowledge and new challenges to accepted truths are arrived at by radically changing one's perspective. However, because it is impossible to genuinely change one's perspective at will,

---

[11] See for example Brown (1987, p. 19).

[12] This is the line taken by most interpreters of Laputa since the publication of Nicholson and Mohler's (1937a, 1937b) influential and Merton's (1966) interesting follow-up.

[13] This is Dutton Kearney's main claim with respect to Book Three in his Introduction (pp. xiv–xv) and comments to the Ignatius Press 2010 edition of *Gulliver's Travels* (Swift 2010).

[14] This is the line taken by Eddy's (1923) influential who writes of the visit to Laputa: "There seems to be no motive for the story beyond a pointless and not too artfully contrived satire on mathematicians . . . . For this attack on theoretical science I can find no literary source or analogue, and conclude that it must have been inspired by one of Swift's literary idiosyncrasies. Attempts have been made to detect allusions to the work of Newton and other contemporary scientists, but these, however successful, cannot greatly increase for us the slight importance of the satire on Laputa." (p. 75). See also Kearney (ibid.).

[15] Elkana (1978).

the required change of perspective is accomplished by leaving oneself intact, as it were, while imaginatively changing reality itself!

It is widely agreed that Lilliput represents the Great Britain of Swift's day, its capital Mildendo, London, and the neighboring Island, Blefusco, France. They are both ruled by emperors, and their government structure, functionaries, and political policies are similar. They share a religiously binding scripture, Blundecral, as do their two real-life Christian counterparts, which they have now come to interpret differently, just as they do. Little is told of Blefusco, but the detailed account of the bi-party structure of Lilliput's parliamentary politics and the history of her domestic and international divides and tensions, rebellions, and pacts adhere unmistakably to early eighteenth century England. All except for one, seemingly immaterial detail: Everything is contracted to a twelfth of its normal size—everything, that is, except for Gulliver, our representative and witness to the unfolding drama.

The effect is astonishing. Attempts to confront, say, the dangers of doctrinal divide by trivializing them (such as likening them to the Lilliput-Blefusco egg-end dispute—a jab Swift seems unable to resist), can achieve little, since the irony will be lost on the disputing parties. But by shrinking the *size* of the people and their habitats Swift succeeds in belittling the great central issues of British politics and reducing them to unimportance, as one critic has it, without having to resort to cynical ridicule.[16] With the exception of the occasional jab, like the egg-end dispute, in narrating the first, second, and fourth voyages, Swift steers wide of the type of lampooning he engages in Laputa and Balninarby, whose aim is not to test, but to deride and ridicule. The three other voyages are elaborate, carefully contrived and controlled thought *experiments*, in which one crucial yet seemingly insignificant factor is changed to render an avidly counterfactual scenario factually enlightening in the extreme!

No measure of reliable empirical data or exact mathematization could begin to yield what is learned about real-life England by imagining Lilliput. Collecting and formalizing empirical data is powerless to challenge and change the framework assumptions by which we amass and interpret them. Uninterpreted data and structures, to paraphrase Kant's famous dictum, are blind and unapplied; the conceptual framework that lends them meaning is empty. Once interpreted and rendered meaningful, the business of what Kuhn dubs "normal science" can begin, and different theories and hypotheses will be proposed and tested against them, leaving the framework itself, the fundamental perspective from which both fact and theory acquire their meaning, unchallenged in principle.

Here lies the power of the counterfactual scenario, or rather, that of scenarios whose diversion from the facts turns out to be of momentous significance despite appearing wholly inconsequential from our initial perspective. By imagining London scaled down unaltered to a twelfth of its size (and ourselves looking on from our original heights), Swift succeeds in throwing the pettiness, stupidity, selfishness, irresponsibility, recklessness, greed, and corruption of British politics into sharp relief starker than could ever be achieved by studying it directly. Making the players smaller doesn't *add* to the pettiness, stupidity or selfishness of their full-size counterparts. It doesn't make them worse. What it does do is to create *distance* between us and them and, hence, between ourselves and our judgment of them; a measure of estrangement that allows us to observe them anew and from afar, as it were. This can only be achieved, as in real experiments, by witnessing a played out scenario, which, since imagined, must take the form of a *narrative* narrated by a trustworthy observer like Gulliver, who stands in for us.

And the same goes for Gulliver's other two voyages, in which the work of penetrating social, normative, and political critique is done, again, by carefully controlled thought experiment—this time, however, with hardly any resort to ridicule or mockery. The seemingly irrelevant size difference, so effectively exploited in Lilliput, is reversed in Brobdingnag, which enables Gulliver, along with his normal size readers, to develop an otherwise unobtainable critical self-perspective on two important normative issues. Terrified in the company of the Brobdingnagian giants and treated most of the time

---

16　Swift (1970, p. 31).

as a not-quite-human curiosity, his self-perception remains, nonetheless, confidently intact. If Lilliput enabled creating critical distance from our taken-for-granted norms of political responsibility by proving them unexpectedly to be proportional to size[17], Brobdingnag did the same with regard to both our aesthetic norms and those of state security. The beauty and erotic allure of the young giant women Gulliver encountered there was hopelessly lost on him, as their size served to revoltingly magnify their faults and blemishes but none of their appealing qualities. The more, it turns out, is by no means the better, in this regard. And the Brobdingnagian monarch's contemptuous reaction to Gulliver's boastful account of the invention of gun powder and the size the British army has a similarly transformative normative impact on our unquestioned assumption that the greater a country's size and strength, the greater is its need for military power and state security.

In all these regards, Gulliver's fourth and last voyage to the Land of the Houyhnhnms, tops the bill. But there is more. The last of Swift's thought experiments teaches us a two-fold lesson: One, similar to the other two, regarding human flaws and shortcomings, the other, intriguingly, about thought experiments themselves. This time, human beings are left at their normal size. The departure from reality consists in a ruling race of intelligent horses more benign, more rational, and more responsible than their human vassals called Yahoos. Houyhnhnms lack any form of religion, and their morality rests on reason alone and its defense. Their attitude is logical, remote, and calculated. They regard death as natural and do not grieve the dead. They practice eugenics and strict family control, and they have no word for "lie". They live peacefully and find war incomprehensibly barbaric. In this regard they are not presented as more human than human *simpliciter* but as better endowed, perhaps, while seriously committed to a different moral code.

Under such circumstances, one would have perhaps expected the Yahoos, the human inhabitants of the land, to have been challenged by the Houyhnhnmians to either deepen and enrich their different way of life, morality and culture, or to adopt theirs, as Gulliver himself seems to have reacted. But by the time of Gulliver's arrival, the Houyhnhnms' presence has had the opposite effect. Rather than rise to the normative challenge, and use the Houyhnhnms' example in normative self-correction, life under the intelligent horses has reduced the human Yahoos to savage brutes exhibiting all the faults and none of the virtues of humankind.

At the most straightforward level, the counterfactual imagined human existence under the command of a master-race superior in intelligence and reliability serves dramatically, even more than in the two other voyages, to expose the faults, frailties and normative fragility of human life and self-esteem. But at a deeper level, it is also a fascinating self-reflection on the limits of the thought experiment itself. Gulliver is the first normal-size human to visit Lilliput and Brobdingnag. What is put to the test of the strange and troubling environments they offer is the well-formed value-system and normative perspective of an intelligent, level-headed, and confident Englishman. But in the course of his final voyage, in addition to the normatively challenging Houyhnhnms, Gulliver encounters a miserable race of humans, robbed of any pretense of normativity and self-esteem by generations of humiliating docile servitude. While Gulliver's outlook is profoundly enriched by each of his voyages and utterly transformed by the fourth, the human-like Yahoos, lacking a robust normative perspective of their own, gain nothing from their day-by-day encounters with their Houyhnhnmian masters except for envy. The normative distance is too great to breach even critically. But, for the same reason, even Gulliver's example can have no redeeming effect. For a thought experiment to be effective, its counterfactual component must be allowed to divert from reality significantly enough to make a real difference but to remain close enough for the perspective it challenges to remain relevant. (This, one might say, is the great difference between *Gulliver's Travels* and those of *Alice in Wonderland* and *Through the Looking Glass*. But that would be the topic for another paper)

---

[17]　Which also characterized the opinion the Brobdingnagian king formed of Gulliver's England, deeming the English "to be the most pernicious race of little odious vermin that nature ever suffered to crawl upon the surface of the earth." (Bk II, chp. vi).

A final word on Newtonian physics: If there is any truth to the reading of Swift outlined above, then his bitter critique of Cartesian rationalism and Baconian empiricism's pretension to provide modern science with adequate grounding and methodology presented in Book III is answered by Gulliver's other three voyages, despite the fact that all three steer wide of science proper. And yet, they are highly relevant to science. Newton's physics rest and derive complete from his three famous laws, or axioms of motion, of which the first—the so-called law of inertia—is the most fundamental[18]. Still, though the idea that the velocity of bodies subject to no external force will remain forever constant[19] forms the very basis for his great work, it cannot be proved empirically nor by mathematical deduction[20]. Galileo's thought experiments were centrally instrumental for the reception of the law and the deep conviction it came to command. It is by imagining the wholly counterfactual scenario of a frictionless boat pushed off onto a perfectly smooth and frictionless lake with no wind or air resistance that the law proves irrefutable.

As I said at the outset, Swift strikes me as the first person to claim that the normal methods of testing belief and opinion for clarity, consistence, coherence, and how they stand to the facts are powerless when applied to deep-seated normative commitments, or what Wittgenstein dubbed "framework truths." To subject our norms to normative critique requires a measure of self-distancing or self-alienation that cannot be achieved merely by looking hard at or thinking hard about our world and ourselves. But by imaginatively changing our world somewhat (or, as I have shown in former work, by exposure to the normative critique of others[21]), such self-estrangement can be achieved with significant transformative results! This is the context in which I find *Gulliver's Travels* a landmark contribution of great philosophical import.

I now turn to the Talmudic literature, in which, as I hope to show, the very same technique is employed in the fields of ethics and civil law to subject basic framework assumptions to the kind normative critique that is impossible to level from within.

## 5. Talmudic Counterfactual Critique[22]

The two *talmudim*, the 5th-century Palestinian Talmud and the 6th-century Babylonian Talmud, offer extensive blow-by-blow engagements with and commentaries on the Mishna, the 3d-century system of Jewish law, *halakha*, on which they expound and whose rulings they often oppose. The lion's-share of their critical halakhic work is done by reinterpretation, which is often radical. But sometimes harsher methods are needed, especially where entrenched normative framework assumptions are involved. Radical reinterpretation of a ruling will be effective only if it remains faithful to what is deemed the law's governing value system. As in science, when, for example, light was deemed to be an undulation rather than a stream of particles, without in any way violating the framework assumptions of Newtonian physics. Attempts to fault and modify elements of the law's normative framework itself require a different approach. And it is at such moments that we find the law exposed to the imagined normative perspective of highly significant others by means of potent narratives. People are not made smaller or bigger in these fictions, nor are animals endowed with superhuman intelligence. Their imaginary, counterfactual element consists in opening halakhic debate to the imagined critique of the civilized gentile world and allowing it to have a transformative impact.

---

[18]　C.f. §5 of the Stanford Encyclopedia of Philosophy entry on Newton's *Philosophiae Naturalis Principia Mathematica*—https://plato.stanford.edu/entries/newton-principia/#NewLawMot.

[19]　Or as Newton himself put it: *"Every body perseveres in its state of being at rest or of moving uniformly straight forward except insofar as it is compelled to change its state by forces impressed."*

[20]　Nor could it be deemed to be self-evident in the usual sense of term. In Aristotle's physics, motion in a straight horizontal line is paradigmatically classed as a form of "forced" change that necessarily requires a motive force. From an Aristotelian perspective, the law of intertia amounted to a contradiction in terms!

[21]　See especially, Fisch and Benbagi (2011, chp. 8) and Fisch (2017a, chps. 2–3).

[22]　The talmudic material analyzed below builds on and away from the latter part of Fisch (2017b). A fuller treatment of the texts, if from a different angle, can be found in Fisch (2019, chp. 4).

Here is a first brief yet potent example. The Jerusalem Talmud, the *Yerushalmi*, relates the following story about the great sage of old, R. Shimon b. Shatah:

> Shimon b. Shatah made a living in the cotton business.
>
> His students said to him, you should take it easier. We shall purchase a donkey for you and you won't have to work so much.
>
> They went and purchased a donkey from a certain Syrian, on which they found hanging a precious stone.
>
> They came and said to him, from now on you need not work at all!
>
> Why?—he asked.
>
> Because we found a precious stone was hanging from the donkey we purchased you from a certain Syrian.
>
> And did he know of it? He asked.
>
> No, they answered.
>
> Go return it to him! He ordered.
>
> Rabbi [Yehuda the Prince] was asked: But did not R. Huna . . . rule in the name of Rav that even those who maintain that one may not steal from a gentile admit that all agree that one is not obliged to return a gentile's lost property [as one is obliged to return that of a Jew]?
>
> To which Rabbi Yehuda firmly responded:
>
> Do you take Shimon son of Shatah for a barbarian?! For he would always prefer to hear 'Blessed be He God of the Jews' than to receive all the riches of the world![23]

The main scriptural source for the law of lost property is Deuteronomy 22:1–3. It is stated with exclusive reference to "thy brother." It is their brethren's stray livestock, and lost property that the children of Israel are commanded to retrieve and go out of their way to return. The word "brother" is repeated five times in the course of the three verses. Whether or not Jewish law at the time recognized gentile property rights in general remains an open question in the story of Rabbi Shimon's donkey. There are rabbis, we are told, who maintained that stealing from gentiles is indeed prohibited, implying that are were others who did not (on this see below). However, the surprised reaction to Rabbi Shimon's insistence that the jewel be returned to its gentile owner (who had not demanded it back, being, it seems, either unaware yet of losing it or unclear as to where it was misplaced) that later prompted Rabbi Yehuda's intervention clearly indicates that until that time[24], the law clearly and unanimously stated that the obligation to return lost property applied only to Jews. This is hence about more than a prosaic first-order ruling, as it involves the very framework distinction between our moral and ethical obligations; between what we owe to fellow humans, and what we owe to our fellow Jews.

What is fascinating about this little yet most significant tale, strategically placed at the heart of the *Yerushalmi's* discussion of the laws of lost property, is, of course, Rabbi Yehuda's response to his disciples' query. But it is also the fact that it is to him that the question is told to have been referred. For Rabbi Yehuda *ha-Nasi*, the Prince is no other than the chief redactor and editor of the Mishna: The person centrally responsible for mishnaic halakha and the highest halakhic authority of his generation. He neither questions nor attempts to reinterpret the objectionable yet bona fide halakhic ruling, nor does he legislate against it. Nor does he try to explain that the saintly Rabbi

---

23  Yerushalmi, *Bava Metzia* 8c.
24  Rabbi Shimon (120–40 BCE) preceded Rabbi Yehuda (135–217 CE) by over two-hundred and fifty years. The implication is, therefore, that it was the latter's firm pronouncement rather than the former's conduct that clinched the normative change. I doubt, however, that in Rabbi Yehuda's day the question of stealing from gentiles was still seriously debated, although, as we shall see, the Tannaitic literature implies that a century or so earlier it was not only debated, but decidedly permitted!

Shimon had decided in this case to go beyond the formal call of duty—all of which would have left it on the books as a viable option of some sort. At times, halakhic duties or prohibitions are suspended for the sake of higher goods, such as "for the sake of peace" or because someone's life is at risk. Rabbi Shimon is not said to have merely *suspended* the questionable ruling. Suspending a ruling due to special circumstances is not to question its propriety per se. Here, however, it is its very propriety that is questioned. The ruling regarding the lost property of a gentile is left on the books as the law, its obnoxious original meaning intact for all to see, but it is declared barbarous—a law indeed, but a law only a barbarian would follow!

To purge a law by moral staining, rather than by conventional legal or judicial methods, is to clearly signal that what is at stake is not merely what the law prohibits or allows but the norm it represents. In terms of religious membership, identity, and obligation, the line dividing Jews and non-Jews is normatively fundamental for both. And it is easy to see how a religious divide could justify and develop into an ethical divide as well, especially given the Torah's widespread use of the terms "brother" (אָחִיר), and "comrade" or "neighbor" (רֵעֶךָ) in stating ethical laws.[25] Yet once such ethical lines have been drawn, their reversal is no longer possible by means solely of internal normative critique. For how can a culture deem its own ethical standards to be unethical if it is by means of those very standards that it makes such judgments?

What is important, therefore, to the present study is the way Rabbi Yehuda elects to *explain* Rabbi Shimon's position to his puzzled disciples (along with many of the readers of this remarkable text). "Barbarian" is not a halakhic or meta-halakhic term. It is an essentially Roman normative category (as opposed to its occasional geographic employment to describe non-Roman countries, as in Yerushalmi, Sukkah, 55b) synonymous with "uncivilized." To ask rhetorically "do you take R. Shimon for a barbarian?!" is to ask whether you take him to be someone *who would be deemed a barbarian* by those who use the word? The use of "barbarian" as an evaluative category in this context conveys an invitation, nay a demand, to imaginatively subject ourselves and our Torah to the critical gaze of the civilized gentile, regardless of religious difference. The giveaway phrase appears in Rabbi Yehuda's last sentence. Rabbi Shimon, it declares, was willing to relinquish all the riches of the world *just to hear people declare "blessed be He God of the Jews"*—"people", as explicitly opposed to "Jews." The moral standard against which the halakhic ethical framework is to be measured is that of the civilized gentile looking in from the outside, rather than that of halakhically practiced Jews judging it from within.

In developing halakha, and especially in the moral critique of existing halakha, the halakhist first obligation is to what, in a different context, as we shall see immediately, the Talmud refers to as "sanctifying God's name"—the term it uses in that context to mean, being forever open to the world, imaginatively attentive to how our laws, our normative choices, and our conduct, appear, sound, and morally resonate outside the boundaries of the halakhic community. To Sanctify God's name is hence to transcend the all-too-cozy, all-too-self-righteous and self-congratulating dialogue conducted by Jewish halakhists among themselves, not by vainly claiming to speak in the name of God's attributes or in the name of the one true timeless moral code of the Torah, but by attempting far more modestly to create a measure of inner distance from what we take for granted by attempting imaginatively to see ourselves as others do, and to try prudently to live up to *their* standards.

## 6. Roman Legal Experts in the House of Study

My second and main example brings all of this out in greater detail and with historical depth. The mishnaic Law of Damages states that owners are responsible for injuries inflicted by their oxen goring the oxen of others. However, in the case of a first offender, as it were, the ox is regarded "innocent," and its owner is charged for only half the damage. Owners of oxen with a history of violence are expected to be more cautious and, in the case of a goring, must pay the full cost. However,

---

[25]    Exodus 20:14 and 22:25; Leviticus 19:13, 17–18, 25:25, 35–36,39; Deuteronomy 19:14, 15:7, 23:25–26.

this applies only to Jews. "Where an ox belonging to an Israelite has gored an ox belonging to a gentile," rules the Mishna, "there is no liability at all, whereas if an ox belonging to a gentile gores an ox belonging to an Israelite, whether *tam* (innocent) or *mu'ad* (with a history), compensation is to be made in full."[26] The Mishna, in other words, imagines a situation in which gentiles are subject to an autonomous system of Jewish law, which, at least since the destruction of the Second Temple, was a counterfactual scenario.

"How can this be?" asks the Bavli in its commentary on this ruling[27]. Either the Law of Damages applies to gentiles or it does not (i.e., either the imagined gentiles under your jurisdiction are counted among your "comrades" or "neighbors" in this respect, or they do not). If it does apply to gentiles, then, in the case of injury to their oxen, they should be compensated like anyone else; if it does not, then just like Temple-owned oxen, they should not be required to compensate others when their oxen are to blame. But you cannot have it both ways, the Bavli implies.[28]

Both *talmudim* start off their discussion of the Mishna's ruling by offering two alternative explanations that are based on earlier sources[29]. The law, they both insist, is indeed not merely inconsistent but blatantly and intentionally unfair to non-Jews. The gentiles, they argue, deserve to be treated unfairly. According to one opinion, their wealth was given to Israel because they failed even to comply with the seven Noahide laws they were given[30]; the other, because they were offered the Torah and refused it. On such readings, the Talmudic fantasy of Jews ruling gentiles comes accompanied by the idea that Jewish law is meant to serve as God's long arm in collectively penalizing the gentiles for having not lived up to His expectations. And at this point, both *talmudim* and, as well as we shall see, the earlier Tannaitic source on which they build, introduce without any forewarning a story about two Roman legal experts who in former times were sent by Rome to assess the Jewish Torah. Here, first, is the Yeruslami's far more direct and far less subtle version:

> It so happened that the [Roman] government sent two legal assessors to learn Torah from Rabban Gamliel[31], and they learnt from him Scripture, Mishna, Talmud, halakha, and aggada[32].
>
> When they had completed their studies they said to him: Your entire Torah is agreeable and admirable except for its following two rulings: [First] that "an Israelite woman should not act as midwife to a gentile woman," yet "a gentile woman may act as midwife to an Israelite woman," [and that] "an Israelite woman should not suckle the child of a gentile, but a gentile woman may suckle the child of an Israelite women in her presence."[33] And [second], that it is forbidden to steal from an Israelite, but stealing from a gentile is allowed.
>
> At that hour, Rabban Gamliel ruled that it was forbidden to steal from a gentile because it is a profanation of the God's name (מפני חילול השם).
>
> As for the ruling concerning the compensation due for goring oxen belonging to Israelites and gentiles, they said, this we shall not report back to the government.

---

26　Mishna, Bava Kama 4: 3.
27　Bavli, Bava Kama 38a.
28　Note that the question is not posed in normative terms but in legalese, as it were—as if the problem with the Mishna's ruling is its inconsistency. It is not its decency or fairness that are questioned. The question's immediate context is the first part of the same Mishna that exempts livestock consecrated to the Temple from the law of damages altogether, both as gorers and gorees, because, as the Mishna explains (ibid), their owner, the Temple does not belong within the category of your "comrades" or "neighbors," as the biblical verse Exodus 21:35–36 has it.
29　C.f. Sifre on Deuteronomy §348.
30　For an authoritative study of the Noahide laws and their history, see Novak (1983).
31　Presumably Rabban Gamliel II of Jabne, who was active immediately after the destruction in 70 CE.
32　Namely, the entire biblical, exegetical, theological, and halakhic corpus.
33　Mishna, Avoda Zara 2:1

Even so, before they arrived at the Ladder of Tyre[34], they forgot everything!

According to the Yerushalmi, the assessors' complaint focused on the general issues of religious discrimination to do with birth and nursing and the basic, essentially ethical issue of right of gentiles to their property. Halakhic permission to rob non-Jews undermines their very capacity for ownership in Jewish eyes and renders everything in their possession legally vulnerable to repossession and exploitation by Jews. However, while the two rabbinic justifications of gentile discrimination clearly reflect unreal fantasies of Jewish political dominion over gentiles, the imperial Roman delegation would have assessed Jewish law more modestly as that of an ethnic community within a broader Roman context. For them, the great threat would have been ethical rather than legal. The Roman legal system applied at the time to all Jewish and non-Jewish inhabitants. There was no immediate danger in their eyes, therefore, for the mistreatment of gentiles by Jewish courts—hence their taking more lightly of the goring oxen ruling. But the two rulings they do disapprovingly quote are deeply disruptive of Jewish-gentile relations regardless of whether the Jewish courts command judicial authority beyond the gates of the community.

This version of the story is significant for three main reasons. First, for the twice-over fictitious,[35] counterfactual scenario of a months-, if not years-long presence of an official Roman delegation engaged in serious study of Torah at a major center of Jewish learning at the feet of a truly great rabbinic sage of a former age; an earlier version our law and way of life under the detailed and meticulous scrutiny of learned gentiles.

Second is the fact that they are described as truly curious about, deeply appreciative of, and in no way antagonistic to Jewish law and learning. Joining in extensive Torah-study renders them members of the learning community even though they are not Jewish. They themselves function in the story as an imagined living refutation of the dismal collectivist view of gentiles as lawless and dismissive of Torah offered jointly by the two preceding rabbinic justifications of gentile discrimination.

Third, for the profound transformative effect their visit and especially, their critique is reported to have had on the ethics of Jew-gentile relations reflected by halakha. I say this because the Yerushalmi clearly implies that had the Roman delegation not won the trust of Rabban Gamliel (as well as that of the latter-day readers of the legend), the deeply discriminatory normative commitment undergirding the two formerly cited rabbinic justifications would have remained firmly in place, and, with it, the legal license to unscrupulously rob gentiles of what they have.

The Yerushalmi tells the story of major normative halakhic reform[36] that would not have been possible but for the imagined exposure of the halakhic system to the trusted scrutiny and critique of idol-worshipping yet civilized gentiles.

What the Yerushalmi, by the end of the day, leaves subtly and un-spokenly unresolved is the mishnaic ruling on which it is commenting. How are we treat the bona fide Tannaitic ruling under discussion that regarding the damage caused or incurred by their oxen, gentiles be rudely discriminated against? The answer, I firmly believe, is obvious. Just as Rabban Gamliel rose to the occasion and ruled the permission to steal from gentiles a profanation of the Name, *hillul Hashem*, the Yerushalmi strongly implies that we too should imagine what this law looks like from a serious gentile perspective and

---

[34]　The ladder of Tyre is Israel's most northern mountain range that reaches the Mediterranean at Rosh Ha-Nikra, about 20 km south of Tyre.

[35]　The story, like all talmudic legends, is related as if authentic, but it is recalled in a work redacted at least three-hundred and fifty years after its supposed occurrence in a context in which its factual credence and accuracy play no role. It is imagined in the sense that the reader is called upon to imagine the resonance of the critique and its transformative impact on Rabban Gamliel and subsequently on an earlier imagined, most likely fictitious, stage of the halakhic system.

[36]　The apparent effect of Rabban Gamliel's related reform (for which this version of the legend is its only source) is found in Tosefta, Bava Kama 10:15, which states that greater is the transgression of stealing from a gentile than from a Jew, because the former involves desecrating the Name!

to firmly follow Rabban Gamliel's example—treat it too as a law never to be followed for the very same reason.[37]

This brief yet potent passage in the Yerushalmi is rendered all the more significant and dramatic when compared to the earlier Tannaitic version of the entire episode from which it clearly sets forth. Its most developed version claims that Deuteronomy 33:3 teaches us that:

> The Holy One blessed be He holds Israel in higher esteem than all the peoples of the world. For so we have learnt: If an ox belonging to an Israelite has gored an ox belonging to a gentile, there is no liability, but if an ox belonging to a gentile gores an ox belonging to an Israelite, there is full liability; an Israelite woman should not act as midwife to a gentile woman, but a gentile woman may act as midwife to an Israelite woman; an Israelite woman should not suckle the child of a gentile, but a gentile woman may suckle the child of an Israelite women in her presence; that it is forbidden to steal from an Israelite, [but allowed to steal from a gentile; and that the lost property of an Israelite must be returned] but not the lost property of a gentile.

In other words, the full list of discriminative laws against gentiles are proudly enumerated as *living proof* of God's favoritism toward Israel.[38] And at this point the story of the two legal assessors in introduced:

> A story was told of two legal assessors who were sent by the Kingdom [of Rome]. They were to told: Go study the Jews' Torah and report its content back to us. They went to Rabban Gamliel at Usha and learned from him midrash, *halakhot* and *aggadot*.

> When their time came to leave they said to him, Our rabbi, your entire Torah is agreeable and admirable except for the following thing[s] that you say: If an ox belonging to an Israelite has gored an ox belonging to a gentile, there is no liability, but if an ox belonging to a gentile gores an ox belonging to an Israelite, there is full liability; an Israelite woman should not act as midwife to a gentile woman, but a gentile woman may act as midwife to an Israelite woman; an Israelite woman should not suckle the child of a gentile, but a gentile woman may suckle the child of an Israelite women in her presence; that it is forbidden to steal from an Israelite, [but allowed to steal from a gentile; and that the lost property of an Israelite must be returned] but not the lost property of a gentile.

> However, we shall not report these things to the authorities.

As we have seen, according to the Yerushalmi's version of the story, Rabban Gamliel's response was "at that hour" to immediately declare the obnoxious laws a profanation of the Name. Here, introduced by the very same phrase, he is said to have done exactly the opposite: To protect the God-willed discriminatory rulings by causing the Roman Assessors to forget them!

> At that hour, Rabban Gamliel prayed that they will remember nothing of this, and they did not.[39]

In normal circumstances (as in *Gulliver's Travels*), imagining how our normative commitments might look to others—especially to those against whom they discriminate—is liable to give rise to one

---

37 The Bavli, (Bava Kama, 37b–38a), though clearly similarly disposed, is unfortunately, to my mind, far too subtle in this respect. In its version of story, the Roman assessors are not described as encountering and prompting the reform of a former stage of halakha in which stealing from gentiles was permitted. Their critique focuses exclusively on the issue of gentile owned oxen, which they declare, before departing, that they will not report to the authorities. The effect is far less dramatic than in the Yerushalmi's version.

38 Reflecting the supposedly earlier stage of halakha, in which stealing from gentiles was permitted.

39 Midrash Tannaim Deuteronomy, 33:3.

of the two responses we have witnessed. It may either prompt us to hunker down in defense of what we self-righteously take God's will to be in the hope that its embarrassing disclosure will be ignored or forgotten, as in the Midrash Tannaim version, or set in motion a self-critical process of normative rethinking and halakhic reform, as in the Yerushalm's use of the very same imagined scenario.

## 7. The Gravest of Sins

In the following Talmudic text, with which I wish to conclude this paper, the imagined normative critique of gentiles looking in is taken explicitly as the ultimate test of personal religious conduct. Bavli, Yoma 86a vividly describes the four main levels of sinfulness and the means available for personal, deontic self-cleansing at each of them. Transgressors of positive commandments need only to repent, and they are forgiven and cleansed of their sins immediately. For transgressors of prohibitions (i.e., negative commandments), however, repentance alone suffices only to suspend their punishment until the Day of Atonement and only then are they forgiven. With respect to transgressions of prohibitions punishable by death, repentance and the Day of Atonement can only suspend punishment, and further suffering completes the atonement. But those guilty of the profanation of God's Name—*hillul Hashem*—"penitence has no power to suspend punishment, nor the Day of Atonement to procure atonement, nor suffering to finish it, but all of them together suffice only to suspend the punishment, and only death itself grants them atonement."

It is a scale of transgressions that culminates in the very gravest of religious offences—worse than murder, worse than adultery, worse even than idolatry—the profanation or desecration of God's name, which is set up sui generis as a category of misconduct to itself, positioned far above all other capital punishment offences. "But what is meant by *hillul Hashem*, in what does it consist," the Bavli then asks. Four different answers are given. The first three are what one would expect.

> Rav said: If I take meat for the butcher and do not pay him at once. . . . R. Yohanan said: In my case [it is a profanation if] I walk four cubits without [uttering words of] Torah or [wearing] tefillin. Isaac, of the School of R. Jannai. said: If one's colleagues are ashamed of his reputation, that constitutes a profanation of the Name . . .

All three opinions associate the desecration of the Name with a sage not living up (or not appearing to live up) to the standards of the community, and acquiring *within* it a bad reputation, which because of his standing, reflects badly on God Himself. All three, however, fail to really explain the religious gravity of such transgressions; what sets them so far above the most gruesome of sins? Abaye might seem at first to be going in the same direction, but the verses he quotes from Isaiah 49 and Ezekiel 36 renders his position quite different from the other three.

> Abaye explained: As it was taught: "And thou shalt love the Lord thy God" (Deuteronomy 6:5), namely, that you should cause the Name of Heaven be loved.[40] If someone who studies and restudies Torah and attends on the disciples of the wise is honest in business and speaks pleasantly to persons, what do people then say of him? "Happy is the father who taught him Torah, happy is the teacher who taught him Torah; woe unto people who have not studied the Torah; for this man has studied the Torah look how fine his ways are, how righteous his deeds!" Of him does Scripture say: And He said unto me: Thou art My servant, Israel, in, whom I will be glorified. (Isaiah 49:3) But if someone studies and restudies Torah, attends on the disciples of the wise but is dishonest in business and discourteous in his relations with people, what do people say about him? "Woe unto him who studied the Torah, woe unto his father who taught him Torah; woe unto his teacher who taught him Torah! This man

---

[40] He achieves this by subtly reading the word *ve-ahavta* (and thou shalt love God) as *ve-ihavta* (and thou shalt render Him beloved).

studied the Torah: Look, how corrupt are his deeds, how ugly his ways"; of him Scripture says: "In that men said of them: These are the people of the Lord, and are gone forth out of His land" (Ezekiel 36:20).

The "people" to whom Abaye refers throughout the passage, as the two cited verses clearly indicate, are not fellow Jews but gentiles observing one's conduct from outside. The verse from Isaiah is addressed explicitly to "the peoples of afar" (the Hebrew actually reads "nations", *le'umim*), and the verse from Ezekiel refers to the "nations" among whom Israel was scattered (36:19). The difference between sanctifying and desecrating God's name has not to do with how well one has studied Torah or how close one adheres to its teaching or one's reputation within the community. It has all to do with how one's conduct impresses the gentiles looking in. Because he is known to be "someone who studies and re-studies Torah, and attends on the disciples of the wise," then if *they* deem his ways to be fine and his deeds righteous, they will come to deeply appreciate the Torah and those who study it. But if they deem his ways to be ugly and his deeds to be corrupt, they will come, for the same reason, to despise Torah and those who learn it. What renders this text so centrally relevant to the present study is that the evaluative vocabulary of the thin normative terms "fine," "righteous," "corrupt," and "ugly," to which Abaye refers is that of the gentile! One commits the gravest transgression of the Jewish religion, according to Abaye, if one's conduct is judged negatively by non-Jews.

The important point, then, is that one cannot afford to wait for them to make their assessment but must do one's best to anticipate their reaction in advance. And this requires all God fearing Jews to constantly and prudently *imagine* how they might be judged by the surrounding gentile population. In other words, at the highest rung of Judaism's scale of normative self-cleansing, one is required to engage in keen and perpetual thought experimentation, imagining oneself normatively scrutinized not by God or one's fellow Jews, but by the imaginary presence of civilized gentiles.

****

Swifts powerful yet implicit claim that in order to subject grounding normative commitments to normative critique, a measure of reflective self-distancing is required that cannot be achieved by direct examination, however intense. Direct examination cannot create such distance. But by imaginatively changing reality and closely examining significantly counterfactual scenarios, we are able to do so, and to great effect. The Talmudic literature, as we have seen, renders Swift's insight a powerful tool for purging the law of meta-halakhic discriminatory bias and one's personal conduct of grave normative shortsightedness.

**Funding:** This research received no external funding.

**Conflicts of Interest:** The author declares no conflict of interest.

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
