# Peer review of "Gulliver and the Rabbis: Counterfactual Truth in Science and the Talmud"

_religions, doi:10.3390/rel10030228_

Round 1

Reviewer 1 Report

See the attached file!

Congrats!

Author Response

Thanks very much for your most positive reading of my text, and helpful suggestions. I have modified the abstract, added a mediating paragraph between the papers two parts, and added a brief summary of the entire argument. 

Reviewer 2 Report

The main thesis of the paper is that thought experiments, namely the close analysis of counterfactual scenarios, “are the only way we challenge normative framework assumptions and learn anything significantly new in and outside science”. In order to argue for this thesis, two main kinds of examples are described. The first kind is Jonathan Swift's Gulliver's Travels. In this case, the A. maintains that what Swift so bitterly satirizes in Laputa and Balnibarbi is not the science of his day, but two foundational philosophies, that is, those of Bacon’s empiricism and Descartes rationalism. In the second part of the paper, the use of counterfactual scenarios is examined with special reference to ethics and law in the Talmud.

There are some problems that should be eliminated:

1)       .Between the first example of Gulliver and those taken from the Talmud, there is a sort of logical gap that should be somehow mitigated. It is advisable to clarify the link between the two parties (along the lines of what is mentioned in 490-496) in the abstract, the introduction, and the conclusion.

2)       As already said, the A. points out that thought experiments, on the basis of a close analysis of counterfactual scenarios, “are the only way we challenge normative framework assumptions and learn anything significantly new in and outside science”. However, in the paper there is no proof of this very strong thesis, according to which there is no other way to challenge normative framework assumptions and learn anything significantly new in and outside science. I suggest to attenuate the formulation of this thesis.

3)    In some of the examples from the Talmud the counterfactual aspect should be explained a little longer. The exposition of the examples could perhaps be a little shortened by resorting to a more frequent use of the indirect discourse, and this would leave more room for considerations that better illustrate the main thesis of the essay. In particular, on line 32, the A. asks the following question: “How can what 32 is patently not the case teach us anything about what it is?” It would be advisable to give more explicitly the answer to this fundamental question concerning thought experiments. In this same order of considerations, it should be noted that the essay is without a conclusion, which instead could summarize what has been said and give better internal consistency to the paper.

4)     The abstract does not reflect the content of the paper in proportion to the arrangement of the treated material and its respective length. The paper requires a more balanced abstract. Regarding the style, another minor shortcoming is that there are too many exclamation points: at least some could be usefully eliminated.

5)     The Author maintains (in  the abstract and in the text) that Jonathan Swift was the first in the modern era who believed that counterfactual scenarios is a reliable way to challenge well-entrenched beliefs, but this is disputable. For example, to mention only one example, Johannes Kepler’s Somnium sive Astronomia lunaris (posthumously published in 1634) might be considered in this way. I would express this point more cautiously: for example, one of the first (most important, etc.) authors.

6)     Some Typos:

Line 15: impressivre / impressive

L. 171 ammass/amass

Line 325 Here, however, it is its very priopriety that is questioned./ Here, however, it is its / very propriety that is questioned.

Line 355: with no wind or air resistence that the law proves irrefutable / with no wind or air resistance that the law proves irrefutable

Footnote 20: , the law of intertia amounted to a contradiction in terms / , the law of inertia amounted to a contradiction in terms

Footnote 21: The talmudic material analysed / The Talmudic material analysed [But the same problem is to be found in many places]

Line 524: reiligious gravity  /  religious gravity

Line 548: withing / within

 Line 552: tro be ugly / to be ugly

Author Response

Thank you for your close reading of my text and helpful criticism.

Most of your comments stem from the false impression I created according to which you take ME to be claiming that thought experiments are the ONLY way to challenge normative framework assumptions. This is the view I attribute to Swift. I myself have spent the last decade or so arguing that the only way to do so is by exposure to the normative critique leveled at us by people committed differently. This is the first time I have considered thought experiments in this context. I have tried to better clarify all of this by modifying the abstract, adding a paragraph explaining the relevance of Swift to the Talmudic material presented in the latter half of the paper, and adding a brief conclusion.

I have also tried to better emphasize that I am not claiming that JS was the first to employ thought experiments, but the first to implicitly claim systematically that with regard to framework assumptions the normal empirical and rationalist methods were to no avail.

As noted, I have tried to better clarify all of this in the new version.

MF